# Characterizing Supply Variability and Operational Challenges in an Intermittent Water Distribution Network

**John J. Erickson [1,2], Yamileth C. Quintero [3] and Kara L. Nelson [1,***

[1]   Department of Civil and Environmental Engineering, University of California, Berkeley, CA 94720, USA; jerickson@hazenandsawyer.com

[2]   Hazen and Sawyer, Dallas, TX 75206, USA

[3]   Instituto de Acueductos y Alcantarillados Nacionales (IDAAN), Panama City, Panama; yquintero@idaan.gob.pa

*   Correspondence: karanelson@berkeley.edu; Tel.: +1-510-643-5023

**Abstract:** Intermittent piped water supply is common in low- and middle-income countries and is inconvenient for users, particularly when supply schedules are unreliable. In this study, supply schedules and operational challenges were characterized in intermittent areas of the Arraiján, Panama distribution network based on one year of pressure and flow monitoring in four study zones, analysis of three years of pipe break data, and observations of system operation. Service quality was found to vary among users and supply schedules were often irregular and unpredictable. Direct causes of unanticipated supply outages included pump failures, chronic pipe breaks in specific parts of the system, transmission main breaks, irregular valve operations, and treatment plant outages. The extent and duration of these outages were often increased by high rates of water loss, insufficient storage capacity, and difficulty detecting and resolving infrastructure failures. Factors associated with intermittent supply, such as intermittent pumping, appeared to be associated with a higher frequency of pipe breaks. However, the analysis did not indicate a strong general correlation between intermittent supply and pipe breaks. Pressure and flow monitoring in intermittent supply areas, similar to that undertaken in this study, could be a valuable tool to improve regular operations as well as longer-term planning and prioritization of system improvements. Water loss reduction and adequate distribution storage capacity could also mitigate the effects of operational failures. Investments in monitoring and data analysis have the potential to improve the reliability of intermittent supply in cases where continuous supply is not immediately feasible.

**Keywords:** intermittent water supply; pressure monitoring; unreliable water supply; pipe breaks; water distribution system; water system operation

---

## 1. Introduction

A water utility operations manager arrives at the regional office in the morning to find a group of local residents waiting to voice their frustration with the lack of water service in their part of Arraiján, Panama. They live near the end of the pipe that supplies their area and normally only receive water intermittently. However, it has now been several days since piped water last arrived. In the outskirts of Arraiján, a rapidly growing peri-urban area outside of Panama City, the area where these residents live was initially served by a rural groundwater system before it was connected to the larger Arraiján water network, operated by Panama's Institute of National Aqueducts and Sewers (IDAAN for initials in Spanish). Since then, the utility has struggled to provide reliable service. Several months before this morning, residents had closed a lane of Panama's largest highway to protest poor service quality.

Today, the operations manager tells the residents he does not know what might be causing problems in their area this time. He pleads with them to be patient while he looks into the matter. This is just one of many fires he needs to put out. He and the regional office are overwhelmed by a backlog of customer complaints and broken pipes in the system that serves a quarter of a million people. All of these problems must be addressed with two active repair crews and one working backhoe.

Further investigation reveals that one of the two pumps serving the area in question is out of service. The one working pump is not enough to push water to the far reaches of the leaky pipe network. Operators had not noticed the problem on their daily drive-by inspections when they listen to check if the pumps are running. The faulty pump is quickly repaired and all is well for a few days until an electrical failure shuts down the entire pump station the next Saturday. The weekend operator does not notice the failure, and the residents again close a lane of the highway before the utility is even aware of the problem.

Utilities in low- and middle-income countries around the world often face problems similar to the ones described above. Intermittent drinking water supply (IWS), defined as supply that is available to consumers less than 24 h per day [1], is a common deficiency in piped water systems [2]. Intermittent supply can be caused by a combination of institutional and technical factors, including the unplanned expansion of the distribution network, insufficient system data to inform an optimal operation of the distribution network, excessive water losses, insufficient water resources, and inadequate infrastructure [3–6]. A recent study [7] extrapolated data from the World Bank Water and Sanitation Program's International Benchmarking Network (IBNET) to estimate that approximately 1 billion people living in low- and middle-income countries worldwide are likely exposed to IWS. IWS is an inconvenience for users [8–10], can make it difficult for a utility to provide equitable supply to all customers in the distribution network [3,11,12], is hypothesized to lead to pipe damage [13–15], and is a risk to water quality [4,16,17]. The nature and severity of IWS varies considerably throughout the world, between water systems, and often within water systems. A recent review revealed the complex and diverse factors that cause IWS and identified that there currently is a knowledge gap regarding characteristics of the different types of IWS and their implications for those involved [15].

While predictable supply is important for customer satisfaction in intermittent systems, many factors make it difficult for water utilities to operate intermittent distribution networks predictably and equitably. High peak demands in intermittent networks often result in excessive pressure losses, as large flows are forced through small-diameter pipes, which lead to supply inequities between users at upstream and downstream ends of a pipe [12]. The operation of intermittent networks is often hindered by incomplete knowledge of the distribution system [3], the inapplicability of hydraulic modeling methods developed for continuous systems [18,19], inadequate monitoring of dynamic hydraulic conditions, frequent pipe breaks [13], and high rates of water loss [6].

Real-time monitoring of intermittent distribution networks is rare and often inadequate when it does exist. Hydraulic conditions in intermittent networks are much more variable than in continuous networks, since pipes may be full or empty during intermittent supply; but agencies managing intermittent networks often do not have SCADA (Supervisory Control And Data Acquisition) or other similar sensor systems to monitor their networks [20]. Even if SCADA is available, monitoring equipment is usually not installed at enough locations to provide a complete picture of complex intermittent networks. In a network studied in Hubli-Dharwad, India, >800 valves were operated during a complete supply cycle, and the state of the system was communicated between employees via phone calls and field visits. Some operating data were recorded in written logbooks, but with varying levels of accuracy [20].

In a previous article, we examined the water quality impacts of intermittent supply in four study zones (one with continuous supply and three with intermittent supply) in Arraiján, Panama [17]. Despite sustained low and negative pressures and water quality sometimes being degraded during the first-flush period (when supply first returned after an outage), random grab samples consistently had good quality. These results contrasted with results reported from a previous study in India, where

water quality in intermittent zones was highly degraded [21], indicating that water quality may vary greatly among intermittent systems depending on the context.

This paper seeks to build on the previous publication to characterize in the same Arraiján, Panama distribution system: (i) the detailed supply schedule experienced by users in the four different supply zones over a 1-year period; (ii) the infrastructural and operational challenges that contributed to irregular and unreliable supply; and (iii) the occurrence of pipe breaks throughout the network over a 3-year period and their relationship to IWS. While previous studies of IWS networks have described supply schedules and inequities based on general observations or surveys of users [21–23], this study provides a much more detailed characterization based on a full year of continuous flow and pressure monitoring in four different study zones. This study also provides novel insights by linking the measured supply patterns to specific operational events and challenges, observed through extensive informal interactions with and interviews of system operators.

For improved administration and operation of complex sectors of distribution networks like the Arraiján study zones, it is important to understand the reality of their current operation. To this end, we characterized supply patterns in the study zones based on their continuity (i.e., the portion of time that water was supplied) and their regularity (i.e., the extent to which the supply schedule followed a consistent pattern). Based on insights from this detailed year-long picture of supply in the four study zones, we identify opportunities to improve service quality in networks facing similar challenges. It should be noted that at the time of publication, many of the supply deficiencies identified herein have not been resolved, despite recent investments to increase conveyance, pumping and storage capacity in Arraiján to address intermittent supply. We hope that the results of this study will help utility managers, policy-makers, infrastructure finance institutions, and researchers to better understand the technical nature, impacts, and causes of IWS with the aim of developing solutions to make supply more continuous, regular and reliable.

## 2. Methods

Data collection for this study was conducted in Arraiján, Panamá from October 2013 to August 2015. Arraiján's drinking water network offered a variety of supply situations to study. Most customers received nearly continuous supply, but others were faced with a range of intermittent supply situations, varying in severity and in how they were controlled. Four study areas were selected in the Arraiján distribution network, each with a different supply situation (one nearly continuous and three intermittent). Supply was monitored in these study zones by a variety of methods, including pressure and flow sensors, field visits, and informal interviews with operators. Pressure and flow sensors provided a more detailed, accurate, and objective characterization of supply than would be obtained by interviewing operators or users. Continuous monitoring over 1 year allowed supply variability and anomalies that may not occur frequently to be captured. Informal interviews with operators allowed us to connect the supply observed to the reality of operating a complex intermittent system. In addition to an in-depth study of the four study zones, pipe break records were reviewed for the entire Arraiján network.

### 2.1. Arraiján's Drinking Water System

Arraiján grew rapidly in recent decades, from 60,000 inhabitants in 1990 to an estimated 263,000 in 2014 [24,25]. This growth varied in terms of formality and the degree of planning involved (Figure 1). The rapid pace of residential expansion made it difficult for IDAAN to expand the distribution system accordingly. Frequent changes in the Arraiján system's management structure (between being managed by IDAAN's central office and a regional office) and difficulties with the implementation of infrastructure improvements also affected IDAAN's ability to respond to the challenges of rapid growth.

Summary statistics on Arraiján's complex water distribution network are provided in Table 1. Despite the ample quantity of water entering the system from the treatment plants, rates of water loss were high and many utility customers in Arraiján received deficient service. The entire system

was vulnerable to temporary outages due to pipe breaks and treatment plant stoppages. Some areas had chronic supply deficiencies caused by: (1) insufficient local distribution capacity (pipe diameter, storage capacity or pump capacity) to supply the water demand in the area; or (2) drawing supply from parts of the network that frequently lost pressure when the capacity of the entire network was surpassed because of high user demand (for example, a Sunday when many users were at home) or operational failures such as pipe breaks. In addition to those with deficient service, some users did not receive piped water at all and were supplied by tanker trucks contracted by the utility.

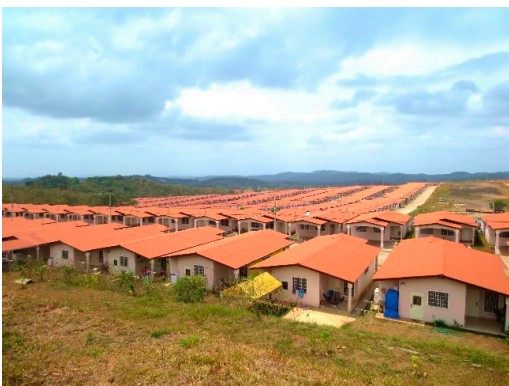 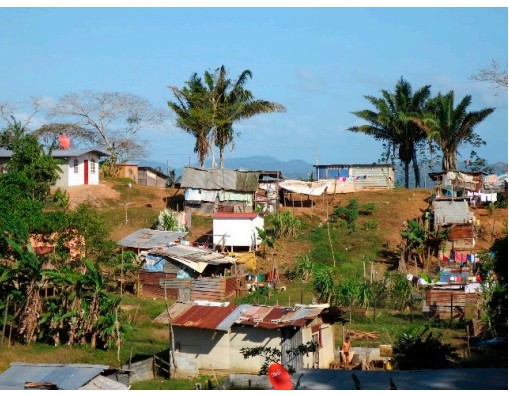

**Figure 1.** Examples of recent residential growth in Arraiján.

**Table 1.** Arraiján Distribution System Vital Statistics (from [26] and interviews with IDAAN personnel).

| | |
|---|---|
| Supply: | • 3 surface water treatment plants supplied by water from the Panama Canal watershed |
| Pipe: | • 431 km PVC, 10-inch or smaller diameter<br>• 73 km ductile iron, 12-inch or larger diameter<br>• Small quantities of cast iron and asbestos-cement<br>• Over half of the network <25 years old. Some portions >35 years old |
| Pump Stations: | • 27, with approximately 3 to 300 horsepower capacity |
| Storage Tanks: | • 39 distribution storage tanks, 38,000–5.7 million liter (ML) capacity<br>• 5.7 ML capacity at water treatment plants<br>• 33 ML total capacity; but 12.3 ML out of service<br>• 20.7 ML available storage = 13.4% of 154 ML daily production * |
| Water balance: | • 154 ML daily production = 585 L/capita (2014)<br>• 310 L/capita not billed to customers = 53% non-revenue water |
| Service quality: | • 6420 connections (13%) received a monthly discount in 2014 due to deficient service<br>• Many more clients suffered occasional interruptions due to pipe breaks or treatment plant stoppages<br>• According to a 2010 survey: 443 households served by tanker trucks because they do not receive piped supply (the number was likely much higher at time of this research, given that 10 trucks were distributing water fulltime) |
| Pipe breaks: | • 604 breaks in pipes ≥ 2-inch diameter in 2014; equivalent to 1.46 breaks/km-year<br>• Similar break rate to the average for 13 Latin American utilities in a regional benchmarking report [27]<br>• Much higher break rate than 0.13–17 breaks/km-year reported in studies of U.S. and Canadian utilities [28] |

* Available storage did not meet IDAAN's standard that storage capacity should equal one-third of daily demand.

## 2.2. Study Zones and Monitoring Locations

The four study zones were selected to be as large as possible, while still maintaining a supply regime with similar characteristics within each zone. The locations of the four study zones in Arraiján are shown in the Supporting Information (Figure S1). The four study zones were located in Burunga, Loma Cova, and Arraiján Cabecera, contiguous sectors within the Arraiján network. These zones were supplied by two of the three treatment plants serving Arraiján, referred to here as WTP A and WTP B. Although the neighborhoods and housing developments in these three sectors varied in terms of urban development and water supply, many shared common characteristics that influenced water supply:

- Complex topography created a need for pump stations.
- The ubiquity of informal housing settlements and unplanned and older (more than 30 years old) developments contributed to the complexity of the water network and often to a lack of data about its configuration.

The four zones are detailed in Table 2. A schematic of Zone 2 is shown in Figure 2 as an example, and schematics of the other study zones are included in the Supporting Information (Figures S2, S3 and S4).

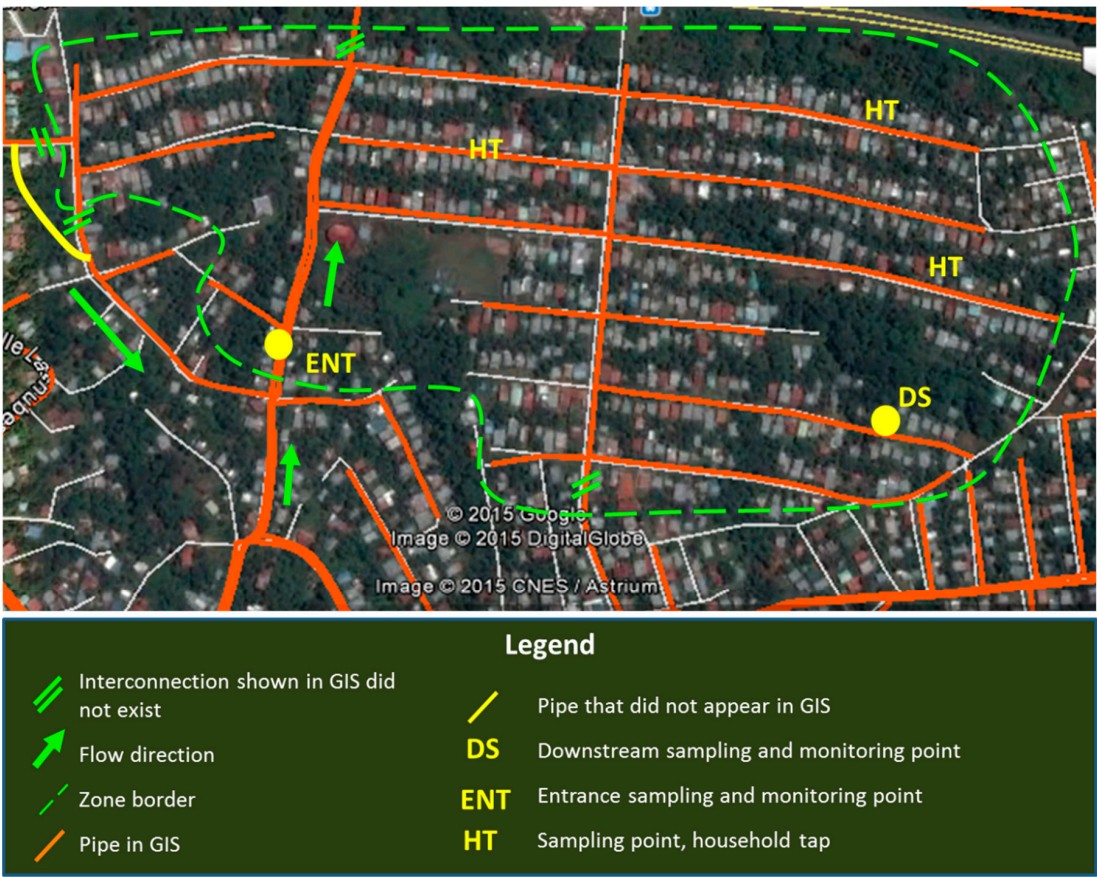

**Figure 2.** Schematic of Study Zone 2. The entrance (ENT) and downstream (DS) locations at which continuous monitoring stations were installed are indicated, as well as the household taps at which water quality grab samples were collected. (Source of satellite images and study zone schematic: Google Earth and IDAAN's GIS database).

**Table 2.** Summary of study zones.

| Zone (Supply Type) | Approx. No. of Customer Connections | Water Source | Supply |
|---|---|---|---|
| 1 (continuous) | 348 | WTP A | Supplied by the main transmission pipe from WTP A through two entrance locations (ENT 1 and ENT 2). Continuous supply except for eleven outages during the year of monitoring and several houses at high elevation. |
| 2 (tank-fed) | 650 | WTP A & WTP B | Received most supply by gravity from two 3.8-million-liter storage tanks and some supply from the main transmission pipe. High elevations lost supply when the storage tanks drained, which was most common during afternoon hours and on weekends. |
| 3 (valve-controlled) | 232 | WTP B | Supplied by a local pump station that supplied water to Zone 3 and two other nearby sectors. Operation schedule called for supplying Zone 3 for three days and then closing a control valve for three days to stop supply to Zone 3 and fill a tank supplying an adjacent area. However, supply often deviated from the schedule due to irregular valve operation, pipe breaks, and pump station failures. |
| 4 (pumped) | 368 | WTP A & WTP B | Most of the supply was from a local pump station pumping directly to the zone's local network. The pump station stopped frequently due to insufficient supply or power failures, causing most of the zone to lose supply. A small amount of supply also entered the zone through two other small diameter pipes, enabling some parts of Zone 4 to have supply even when the main pump station was off. |

*2.3. Continuous Pressure and Flow Monitoring*

Continuous pressure and flow monitoring at the entrance(s) and a downstream location in each study zone allowed detection of when the supply was on and off. Pressure was monitored with ECO-3 RTUs (remote telemetry units, AQUAS Inc., Taipei, Taiwan). The pressure monitors normally recorded a measurement every 30 seconds (s). They also were programmed to record measurements more frequently when a pressure transient was detected. In addition to measuring pressure, the RTUs received signals from other sensors and sent the data periodically to an internet server. The RTUs also had the capacity to send text messages when pressure or other parameters went out of a programmed range. At the Zone 3 entrance monitoring point, pressure was monitored by an LPR-31i pressure monitor (Telog Instruments Inc., Victor, NY, USA). Data were downloaded from that sensor each week to a laptop computer.

IP80 Paddle-wheel insertion flow meters (Seametrics Inc., Kent, WA, USA) were installed at the entrance(s) to each zone. These sent an electrical pulse signal to the RTUs. Some stations were also equipped with turbidity and free chlorine sensors for a related water quality study (results reported in [17]).

Monitoring equipment was installed in above-ground metal boxes (Figure 3). Each set of equipment was powered by a 12-volt battery charged with a solar panel installed on the top of the box. Each monitoring station was connected to the distribution pipe via a saddle installed on the pipe, a $\frac{1}{2}''$ PVC pipe, and a 3/8″ PVC hose.

Pressure data were smoothed (running average of five nearest data points) before analysis. Zones 1–3 were considered to be without supply when pressure (at ground level) was <2 psi at the downstream monitoring station. Zone 4 (pump-controlled) was considered to be without supply when the Zone 4 pump station was stopped (with both pumps off) because the Zone 4 downstream monitoring station received supply from interconnections with adjacent areas of the network and often had supply even though the pump station was off and much of Zone 4 was without supply. Outages with less than 10 minutes (min) of supply between them were grouped together and considered single outages for analysis purposes, but reported durations only include the time when water was actually off. Outage groups with total duration < 10 min are not reported.

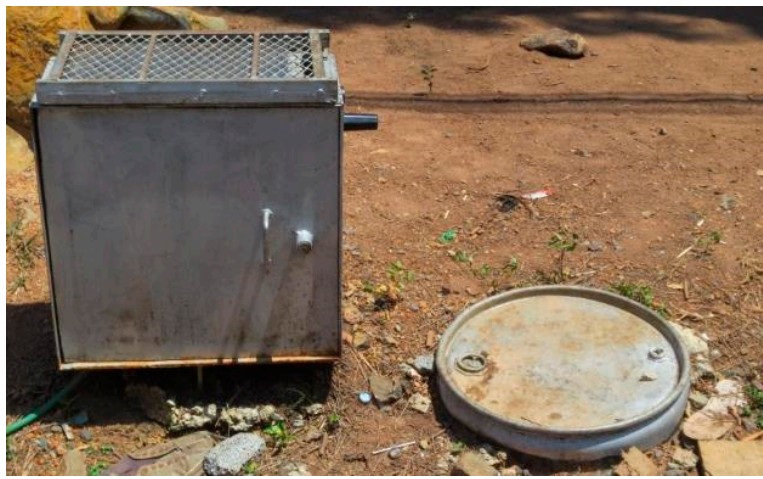

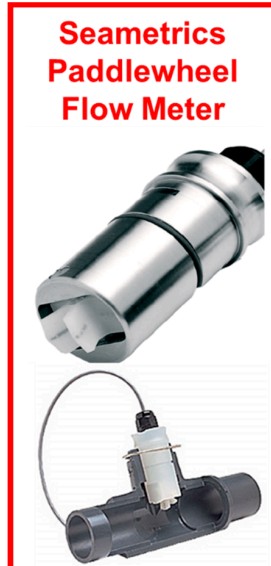

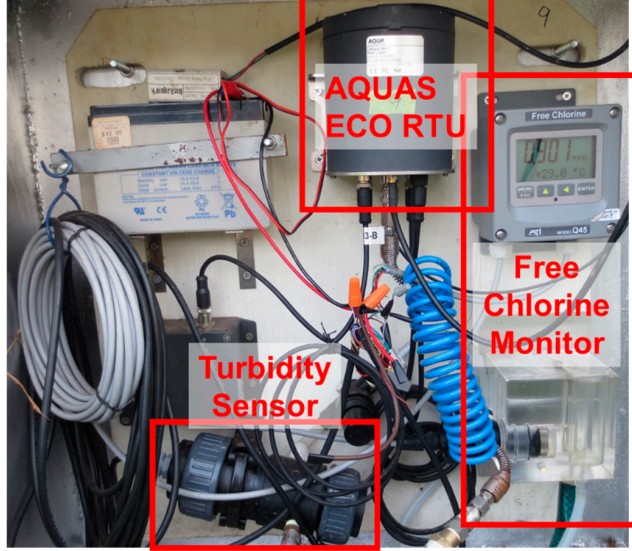

**Figure 3.** Continuous monitoring station (**top**) and sensors (**bottom**).

*2.4. Qualitative Observation of Water Supply Schedule*

To complement the data from continuous monitoring stations, observations were made at three household taps in each zone whenever grab samples for water quality analysis were collected. This sampling provided the opportunity to observe supply conditions in parts of the study zones that did not have continuous monitoring stations. Each time a household tap sample was collected, the researcher asked the user whether they had experienced any supply interruptions in the last week and, if so, when the last interruption had ended. Sample households were distributed geographically throughout each zone, so qualitative observation of supply in these locations during the approximately 500 household sampling visits enabled a more complete picture of supply in the zones.

This portion of the research was carried out under Protocol 2012-04-4278, approved by the UC Berkeley Committee for Protection of Human Subjects on 17 June 2013.

*2.5. Qualitative Observations of Network Operation*

Hundreds of hours were spent informally observing and interacting with Arraiján system operators, which offered an up-close view of the operation of the network. Those informal observations, when coupled with continuous monitoring data, provided insight into the challenges of operating a complex intermittent distribution network. When hydraulic events of interest were captured by

continuous monitoring, system operators were interviewed informally to better understand what had occurred. Also, when operators mentioned problems in the study zones, the relevant continuous monitoring data were reviewed. This back and forth between operators' observations and hydraulic monitoring data also permitted an assessment of whether and how such hydraulic monitoring might be useful to operators.

*2.6. Pipe-Break Analysis*

Pipe break repair records for the entire Arraiján network during 2012–2014 were analyzed to compare break rates in different parts of the network. Records were analyzed to identify areas of the network with particularly high break rates and assess whether there was an association between the frequency of pipe breaks and intermittent supply. Although these records represented repairs instead of breaks, for this analysis each repair is referred to as a break. Based on the location written on the form filled out by the repair crew, each break was assigned to a zone (a neighborhood or housing development). The length of the pipe in each zone was calculated using the utility's GIS database. Pipes with <2-inch or >12-inch diameter were excluded from the analysis. (The small-diameter pipes were not in the GIS database, so could not be included. The large-diameter pipes were normally transmission pipes and the pressure regime in those pipes was often not related to the supply regime in the zones they passed through.) Some zones of Arraiján for which pipe information was not available in the GIS database were excluded from the analysis.

To categorize supply continuity and the approximate age of the pipes in each zone, the utility's field supervisor, who had more than 30 years of experience working for the utility in Arraiján, was consulted. Pipe break data were analyzed in R [29]. Statistical tests for independence were done with permutation tests using the coin package [30] because these do not require assumptions regarding the distribution of the data. The threshold for statistical significance was $p < 0.05$.

## 3. Results and Discussion

*3.1. Heterogeneous and Irregular Supply in Study Zones*

Results from approximately one year of pressure monitoring revealed that supply continuity varied widely between the study zones and that supply in the intermittent study zones was often highly irregular.

### 3.1.1. Supply Schedule in Study Zones

Supply statistics for each study zone, based on continuous monitoring data, are summarized in Table 3. Zone 3 (valve controlled) was without water for the largest fraction of monitoring time (43%), as would be expected given the utility's plan for supply to be on for 3 days and then off for 3 days, followed by Zone 2 (17%), Zone 4 (13%), and Zone 1 (0.9%).

**Table 3.** Summary statistics of supply in each zone. Supply was considered to be off in Zones 1–3 when the pressure at the downstream station was <2 psi at ground level. Zone 4 was considered to be without supply when the pump station serving the zone was off. Average pressure for all zones is at the downstream monitoring station.

| Study Zone | Zone 1 (Continuous) | Zone 2 (Tank-fed) | Zone 3 (Valve) | Zone 4 (Pumped) |
|---|---|---|---|---|
| Monitoring time (days) | 350 | 318 | 317 | 349 |
| Time without supply (days) | 3.2 | 54.5 | 137 | 47 |
| Fraction of monitoring time without supply | 0.9% | 17% | 43% | 13% |
| Average pressure when there was supply (psi) | 22 | 38 | 36 | 47 |
| Number of supply outages | 11 | 107 | 114 | 336 |

The distributions of outage durations at each of the downstream monitoring stations are shown in Figure 4. The tank-fed zone (Zone 2) had 107 outages lasting up to 3 days. A typical outage began

during the afternoon when the upstream storage tanks serving Zone 2 drained, and ended around midnight once the level in the tanks recovered. Longer interruptions occurred when the upstream storage tanks were without water for longer because of a supply deficit in the overall network caused by a pipe break or pump or treatment plant shutdown.

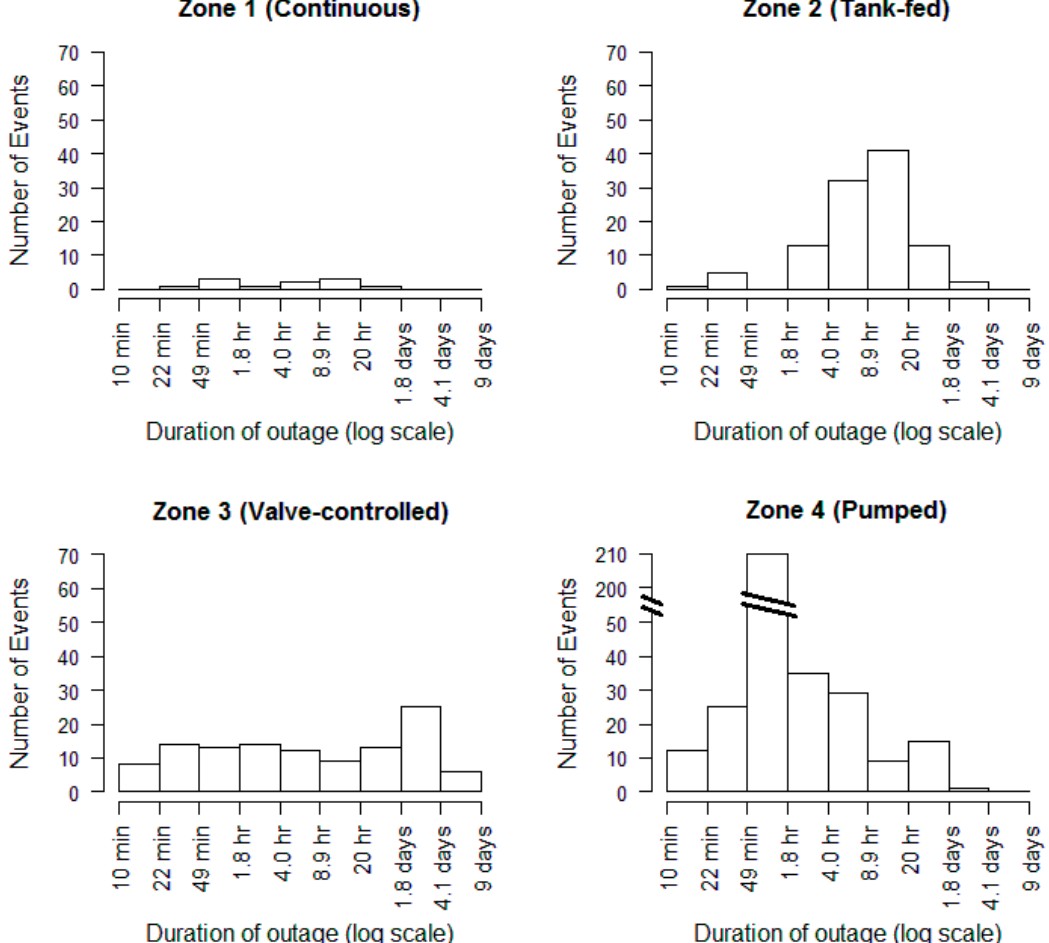

**Figure 4.** Distribution of outage durations for each study zone. Note that the x-axis is a log scale with nine evenly spaced bins between 10 min and 9 days; the y-axis for the Zone 4 graph is cut.

According to the operation plan for the valve-controlled zone (Zone 3), outages there should have lasted 3 days. While the most common outage length was about three days (22% of outages were between 1.8 and 4.1 days long), many measured interruptions were much longer or shorter. Eight interruptions of >4 days occurred. The three longest, lasting 6.3, 6.6, and 8.2 days, were associated with breaks in the single 4-inch pipe conveying water to Zone 3. Many shorter interruptions occurred when the Zone 3 pump station stopped temporarily during a supply period or the valve at the entrance was left partially open and supply at the downstream monitoring station fluctuated between off and on depending on demand elsewhere in Zone 3. (Operators sometimes left the valve at the entrance to Zone 3 partially open when it was scheduled to be closed so some flow entered Zone 3 and the rest went to a storage tank in an adjacent sector. This strategy was employed to supply the adjacent sector but avoid overflowing the storage tank there. During these times when the valve was partially open, the lower elevation portions of Zone 3 had supply and the higher portions did not, with the boundary between households with supply and without supply moving as demand fluctuated. Supply at the downstream monitoring station fluctuated between on and off during these times as the supply boundary moved back and forth across the monitoring location.)

Outage durations in the pumped zone (Zone 4) varied widely, since the pump station was not run according to a schedule, and stopped whenever the suction tank that it pumped from emptied or the electricity supply was interrupted. Seventy percent of the 336 outages, representing 24% of the time that the pump station was off, lasted between 30 and 120 min, likely because this was the approximate length of time it took for the suction tank to fill from the level at which the pump shut off to the level at which it turned back on. Ten outages lasted more than 24 h, with the longest one lasting 48 h. Between 29 April and 7 May 2015, the Zone 4 pump station stopped daily at 8:50 p.m. and started again at 4:52 a.m., causing the incident described in the introduction. During that time, one of the pumps was damaged without operators being aware of it and the other was programmed to stop during the night.

Data were also analyzed to determine whether outages were more common at certain times, days, or seasons. In Figure 5, the percentage of time during each hour of the day and each day of the week that each zone was without supply is shown. A high percentage means that the zone was without water more frequently during that hour of the day or day of the week. Service continuity did not vary noticeably by the hour of the day or day of the week in Zones 1 and 3. In Zone 1, the percentage of time that water was off was low at all times. In Zone 3, the valve was typically open or closed for several days at a time, which meant that there were no specific hours or days during which the zone was without service. On the other hand, supply in Zones 2 and 4 varied noticeably by the hour of the day and Zone 2 supply also appeared to vary by day of the week. In Zone 2, supply was off more often between 3 p.m. and 10 p.m., and on weekends, when it was more likely for the upstream storage tanks supplying the zone to be empty after being depleted by higher daytime demand. In Zone 4, supply was off more often between 8 a.m. and 5 p.m., probably because during those hours demand was higher in other parts of the network, which reduced system pressure and reduced supply to the Zone 4 pump station.

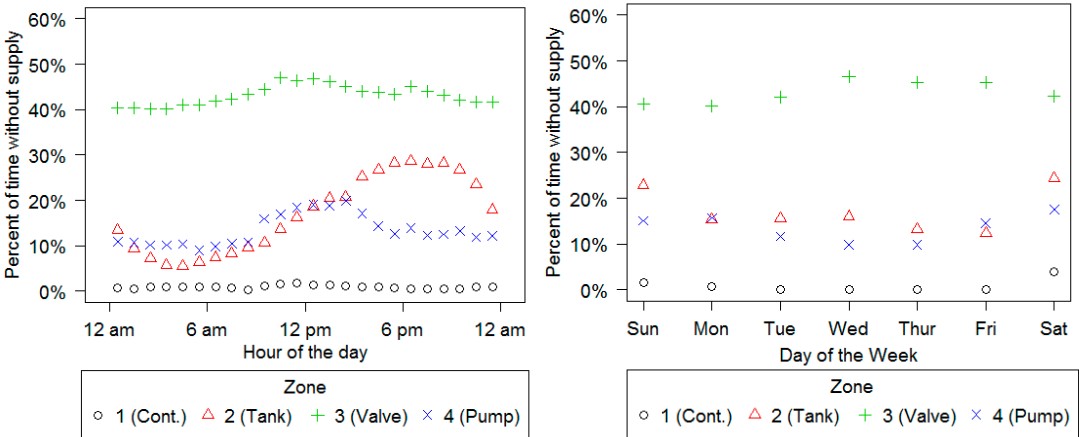

**Figure 5.** Percent of time each study zone was without supply by the hour of the day and day of the week.

In Figure 6, the variation in the percent of time that each zone was without supply throughout the year is shown. On some occasions, supply problems occurred in one zone and the other zones were unaffected. For example, at the end of October and beginning of November, Zone 3 was affected by two long outages (8.2 and 6.4 days) associated with breaks in the 4-inch pipe supplying that zone, but supply remained typical in the other zones. On other occasions, large-scale supply problems affected all four zones at once. For example, during the beginning of September and the end of January, all four zones were affected by breaks in the 24-inch transmission pipe from WTP A.

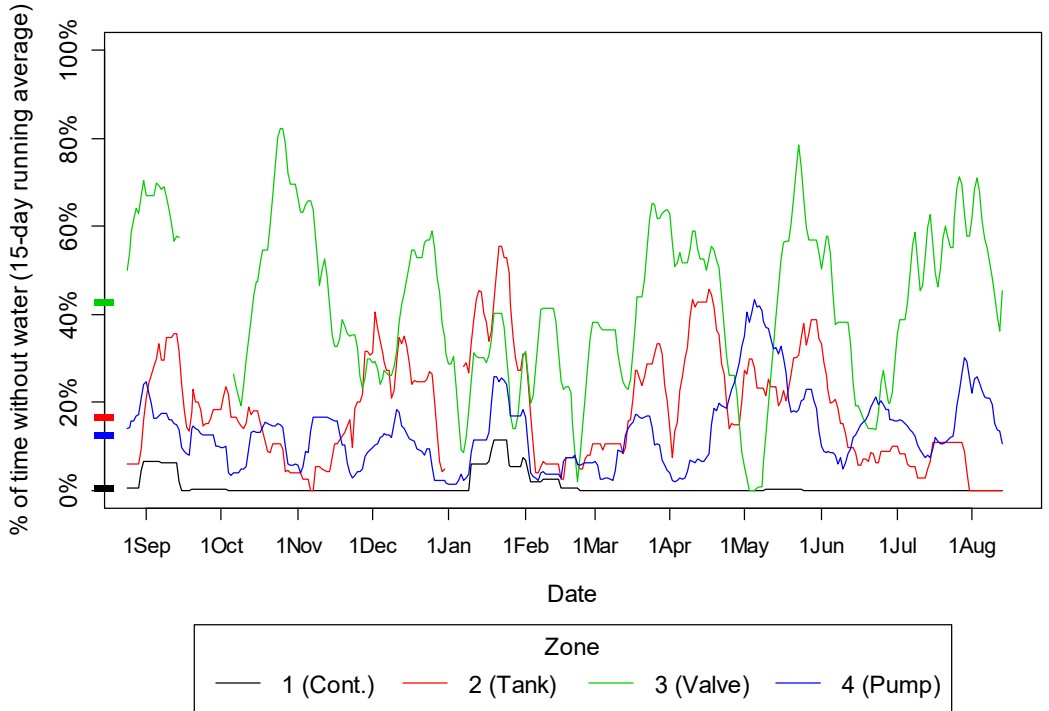

**Figure 6.** Fifteen-day running average of the percent of time that supply was off in each zone. For each day, the average percent of time water was off during the 15 nearest days is shown. Values are only shown if at least 7 days (out of 15) of data were available. Colored dashes on the y-axis mark the average percent of time water was off in each zone.

### 3.1.2. Variable Service Quality across the Arraiján Network

As can be seen from the data presented above, there were marked disparities in service quality across the four zones studied within the same distribution network. Supply ranged from nearly continuous in Zone 1 to very intermittent and prone to extended outages in Zone 3. It should be noted that the four study zones were not chosen to capture the full range of service quality found in the Arraiján network, but rather to focus on the different intermittent supply regimes found in the network. Much of the network had supply equally or more continuous than Zone 1, and, as mentioned in Section 2.1, portions of Arraiján received no piped supply at all and were supplied by tanker trucks. Thus, disparities across Arraiján as a whole were even wider than the disparities seen across the study zones.

While Section 3.1.1 describes approximate supply conditions in each zone based on conditions at the downstream monitoring station (Zones 1–3) or on whether the pump station serving the zone was on (Zone 4), supply also varied geographically within each zone based on elevation and distance from the zone's entrance. For instance, walking 50 yards up a hill in Zone 1 could take you from a house where supply rarely went out to a house where supply went out most afternoons. Spatial variation in supply continuity resulted in service quality being unequal between neighborhoods and even between neighbors.

Previous research has identified uneven service quality as a common problem with IWS [3,11,12] and provided evidence of such inequities in IWS networks [22]. Our continuous monitoring data from Arraiján supports those findings and provides a more detailed picture of uneven service quality.

### 3.1.3. Irregular and Unreliable Supply

Galaitsi et al. [15] proposed classifying IWS into Predictable Intermittency ("shut-offs that occur generally according to a predictable and anticipated schedule"), Irregular Intermittency ("supply arriving at unknown intervals within short time periods of no more than a few days"), and Unreliable

Intermittency ("uncertain delivery time and risk of insufficient water quantity, often exacerbated by limited storage and long periods of non-delivery").

Pressure data demonstrated that supply in all four Arraiján study zones had elements of predictability, irregularity, and unreliability, depending on the time scale considered. While supply in each zone cannot be strictly classified into one of the categories proposed by Galaitsi et al. [15], the categories are still useful for describing supply in each zone. Zone 1 supply was normally predictably continuous but did include a few unanticipated outages lasting up to 22 h. For a user accustomed to continuous supply and unlikely to have a large volume of water stored, an outage of 22 h may be perceived as unreliable. Supply in Zone 3 was intended to follow a predictable three-day-on, three-day-off schedule, but in practice, this zone often had the most unreliable supply, with unexpected extended outages. Zones 2 and 4 normally had irregular supply, with typical outages being short enough to be manageable for users. However, they also experienced bouts of unreliability when infrastructure failures caused more extended outages. As mentioned in Section 3.1.2, supply also varied within zones, and some users experienced a much more irregular and unreliable supply than that registered by our pressure monitoring.

Unreliable supply is an inconvenience and hardship for users. Burt and Ray [31] argue that customers' satisfaction with water supply is affected by quantity, quality, convenience, and reliability. Reliability is a particular concern in intermittent systems since supply often comes at irregular times and is unpredictable. In an intermittent system in Hubli-Dharwad, India, users who had never experienced continuous supply placed more value on punctual supply, increased frequency and duration of supply, and water quality than they did on receiving continuous supply [32]. Unpredictable supply can disproportionately affect lower-income households if they are less able to mitigate its effects. For example, in their study of intermittent supply in Hubli-Dharwad, India, Kumpel et al. [33] found that households that did not have rooftop tanks and used a public tap or a neighbor's tap shared with more than 2 households on average used only approximately 20 L per capita per day, a quantity likely insufficient for household uses such as bathing and laundry [34]. As will be discussed further in Section 3.4, irregular and unpredictable supply conditions also made the operation of the Arraiján network difficult for the utility.

*3.2. Pipe-break Analysis*

The average pipe break rate across the 142 zones analyzed was 1.42 breaks/km-year. Some breaks were fractioned between multiple zones when the recorded location was not specific enough to know in which of the zones the break occurred. Also, some breaks for which the diameter of the pipe was unknown (and may have been <2 inches and thus should have been excluded) were discounted by 39%, the portion of all breaks that occurred in pipes with diameter < 2 inches.

Break rates varied widely by zone (Figure 7a). In 54 zones there were no recorded pipe breaks during the 3 years. Twenty-three percent of all of the breaks occurred in just ten zones (see Supporting Information Table S1), even though they only had 2.9% of the pipe length. Some small zones (such as the zones ranked 1st and 7th) may have had artificially high pipe break rates either because some pipes in these zones were not registered in the GIS database (such that the total pipe length is actually longer than the value used for analysis) or because some breaks in nearby zones were classified as within these zones. The second-ranked zone included the 6-inch transmission pipe between the Zone 4 pump station and Zone 4. Thirty-nine of the 51 breaks in that zone were in the 6-inch pipe going to Zone 4. Zone 4 itself does not appear in the top-ten list but also had 20 breaks registered on the same 6-inch pipe. The frequent breaks in that pipe are likely due to high transient pressures from the intermittent pumping (see Supporting Information Figure S5). The zone with the third-highest break rate also had a pump station that stopped frequently and had a known problem with pressure surges. The fifth-ranked zone was Study Zone 3, controlled by intermittent valve operations. Study Zone 1 (continuous) ranked 45th, with a break rate of 1.08 breaks per km per year, and Study Zone 2 (tank-fed) ranked 29th, with a rate of 2.16.

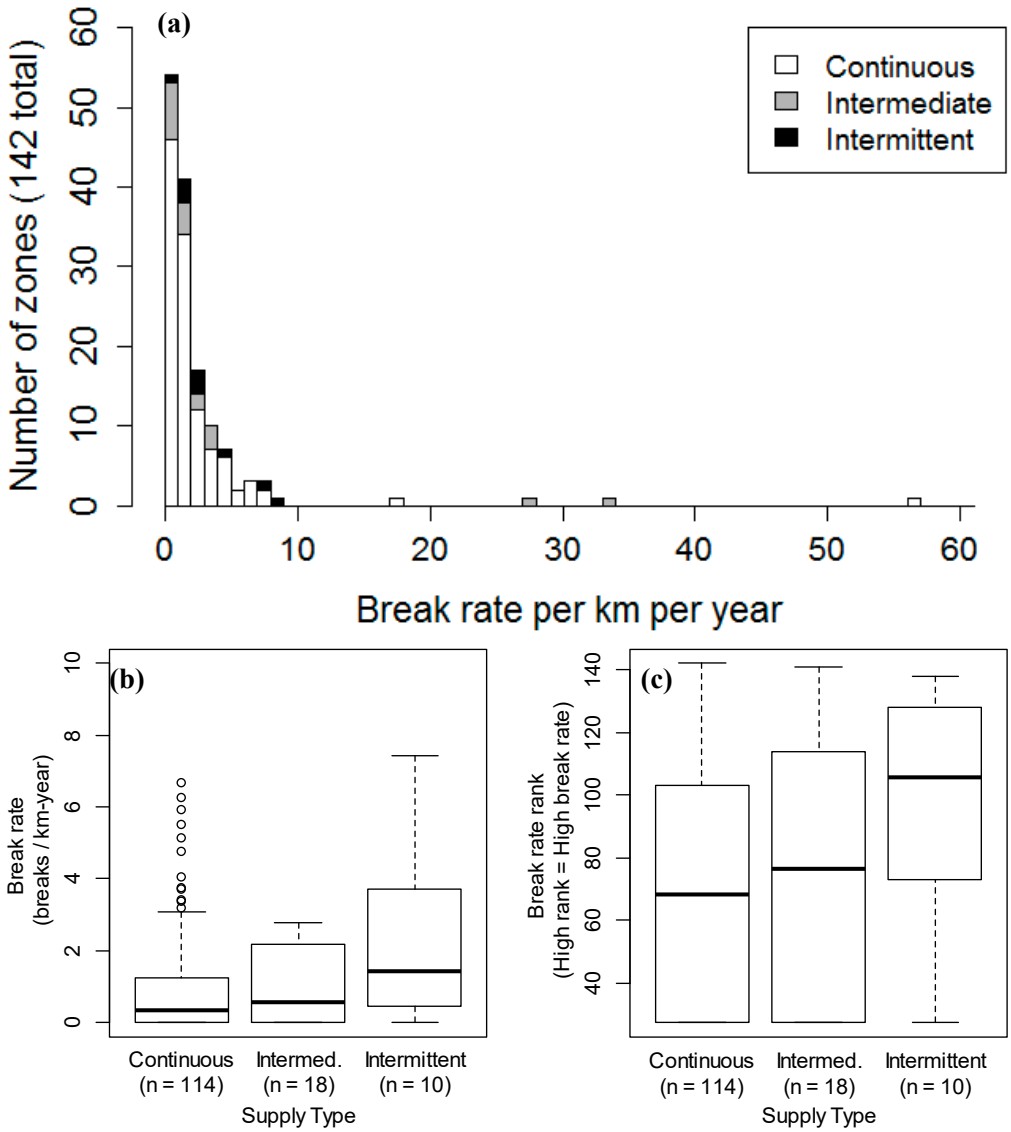

**Figure 7.** Analysis of 2012-14 pipe break data: (**a**) Distribution of break rates for each zone, (**b**) boxplots of break rate by supply type, and (**c**) and break-rate rank by supply type. In plot (**a**), the first bar (on the far left) represents the 54 zones with no breaks. Four zones with break rates > 10 breaks/km-year are excluded from the plot (**b**). Note that the minimum rank in (**c**) is 27.5 because 54 zones had no breaks and were each assigned an average rank.

The high fraction of breaks occurring in a small number of zones, some of them with known sources of transient pressures, suggests that by investigating and improving pressure conditions in a few key parts of the Arraiján network, the utility could drastically reduce the frequency of breaks and the associated supply interruptions and repair costs.

The potential relationship between the break rate and supply continuity was investigated by classifying supply into three categories. "Continuous" supply meant that the zone only lost supply when a large portion of Arraiján was without water; "Intermittent" supply meant that the zone regularly was without water, and; "Intermediate" supply meant that the zone normally had continuous supply but was vulnerable to losing supply when pressure was low in the main network. Break rates and break rate ranks are compared by supply type in Figure 7b,c. In one-way permutation tests for independence (two-tailed tests, classifying supply and age categories as ordered factors, and using break-rate ranks), higher break rates were significantly associated with more intermittent supply ($p = 0.042$) and higher pipe age ($p = 0.030$). In a permutation test for independence where both supply

and pipe age categories were considered simultaneously, the association almost met the threshold for significance ($p = 0.058$).

While the permutation test showed a marginally significant association between more intermittent supply and high break rates, that association might have been driven mainly by a few zones affected by intermittent pumping and not indicate a general effect of IWS. As seen in Figure 7a, some intermittent zones had very low break rates, and some continuous zones had high break rates. Also, in some cases, other factors associated with IWS may be the actual cause of high break rates instead of the hydraulics of intermittent supply. For example, in Study Zone 3, many breaks occurred in a location were the pipe conveying water to the zone was suspended in the air to cross a stream, an installation constructed by the local residents. Intermittent areas, due to their often informal development, may tend to have more contributing factors that lead to pipe breaks.

Although the data did not show a clear-cut correlation between IWS and water main breaks, IWS could have been more strongly associated with service line breaks and leaks, which were not analyzed. A study in Cyprus [35] found that implementation of IWS caused a significant increase in the vulnerability to failure for household service lines but not for larger water mains.

### 3.3. Operational Challenges Contribute to Unreliable Supply

Examples from the study zones illustrate how a lack of information on the current hydraulic state of the distribution system delayed the detection of infrastructure failures and made supply less reliable. In one case, the capacity of a different pump station near the Zone 4 pump station was increased, which reduced flow to the Zone 4 station and reduced supply to Zone 4. The utility did not anticipate these effects, and changes were only made to resolve the situation after Zone 4 residents blocked a lane of Panama's largest highway to protest the decline in service quality (the first highway closure mentioned in the introduction). A second situation in Zone 4, where undetected pump station malfunctions led to poor service quality, user dissatisfaction, and eventually another highway lane closure in protest, was described in the introduction.

These problems with the Zone 4 pump station were all apparent when continuous pressure and flow data collected at the pump station discharge as part of this study were reviewed afterward. If operators had access to such data and monitored it routinely (or set up relevant alarms to alert them of problems), situations like these might be avoided.

In the valve-controlled zone (Zone 3), inconsistent operation and the inability to monitor supply conditions caused actual supply to deviate substantially from the utility's schedule of 3 days on and 3 days off. The control valve was sometimes not operated according to schedule, because operators were not available to open or close it due to another crisis or commitment, or because weekend operators were unaware of the intended schedule. Delay in detecting and repairing breaks in the 4-inch pipe supplying Zone 3 caused the three longest outages. Zone 3 valve operations and pipe breaks were also visible in continuous monitoring data. Such data could help the utility operate the valve more consistently and respond faster to pipe breaks. As one example of the latter, continuous pressure and flow data from the Zone 3 entrance during a pipe break are shown in Figure 8. Flow increased and pressure decreased to a negative value at the time of the break. Approximately 8 h later, the flow stopped and pressure increased to zero when an operator closed the control valve located just upstream of the entrance monitoring station.

The incidents described above from the study zones are indicative of operators' general inability to monitor the network and detect infrastructure failures, such as breaks in distribution pipes or pump malfunctions, before users had already been severely affected. Even though Arraiján's distribution network was quite complex, the utility operated it with little information about its current state. Some of the 27 pump stations frequently malfunctioned. Apart from the monitoring equipment installed for this project, only one of the pump stations could be monitored by telemetry. To monitor the others, operators had to do daily field inspections driving around in a truck.

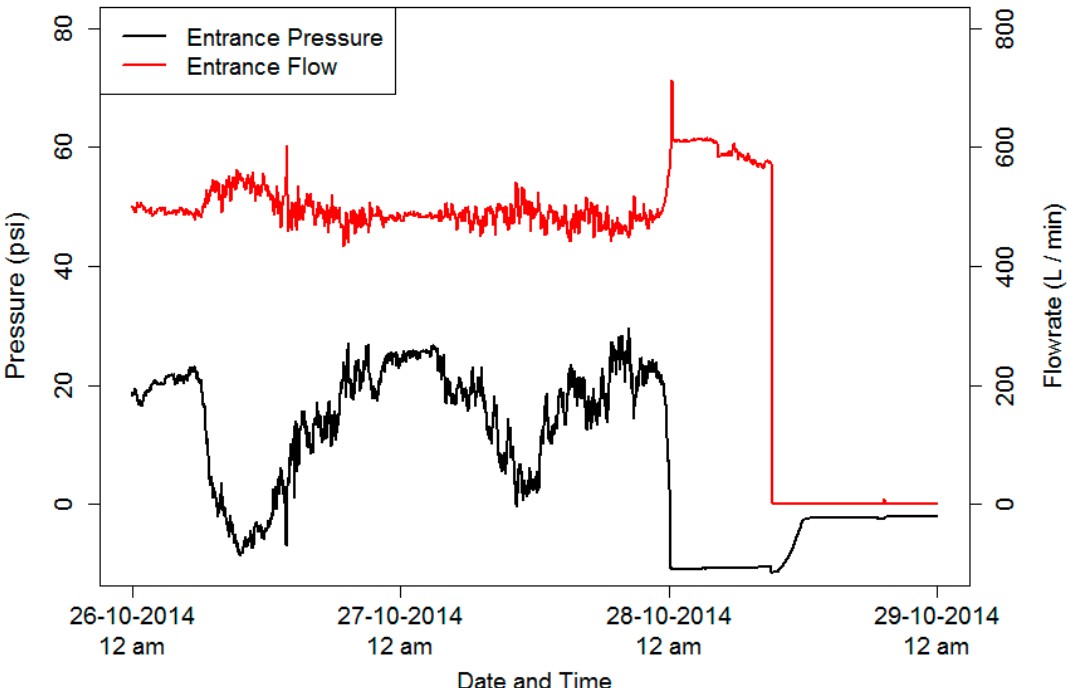

**Figure 8.** Continuous pressure and flow monitoring data from the Zone 3 (valve-controlled) entrance monitoring station during a break in the 4-inch pipe supplying Zone 3 (downstream of the entrance) at approximately 12 a.m. 28 October 2014.

While the localized failures discussed above caused many of the unplanned outages observed in the study zones, other outages observed in the study zones were the result of system-wide deficiencies. Although the Arraiján system had more than enough supply capacity, high rates of water loss, a lack distribution storage capacity, and limited ability to sectorize and control the system prevented the utility from being able to cope with temporary operational failures such as treatment plant stoppages, pump shutdowns, and pipe breaks. According to a log kept by utility managers, between August 2014 and July 2015, users in a large portion of Arraiján were without supply on 13 occasions (eight unexpected and five planned), due to such temporary operational problems.

Limited information on pipe connectivity and a lack of control valves often made these failures difficult to resolve and prolonged their effects. For example, incomplete maps of the pipe network often made it difficult for operators to determine the cause of an outage or to shut off a sector of the system to repair a pipe break. Repair crews lost time trying to determine how to depressurize the sector where a break was, and, due to a lack of control valves, sometimes had to cut off service to a large portion of the network in order to depressurize the area near a break.

*3.4. Opportunities to Improve Service Quality*

Observations from this study suggest that utilities could improve service in intermittent networks like Arraiján's by making both localized and system-wide operational improvements. Locally, intermittent supply could be made more consistent and reliable by using pressure and flow monitoring routinely and/or as a diagnostic tool. System-wide, reducing water losses, providing adequate storage capacity, and improving monitoring could make supply more continuous and reliable.

3.4.1. Local Monitoring for More Reliable and Transparent Intermittent Supply

Continuous hydraulic monitoring in sections of the distribution system with intermittent supply can be used as a diagnostic and tracking tool. As demonstrated in Section 3.1, such monitoring can identify areas where service is most deficient or unreliable. Operators, managers, and planners can then direct attention, resources, and capital improvements to those areas. Once corrective action has

been taken, the same monitoring methods could be used to evaluate whether the measures taken resulted in the intended improvements, such as a more reliable supply schedule. Monitoring results could be made publically available to increase transparency and make users aware that the utility is working to identify and address supply problems.

In addition to being a tool for identifying chronic problems and prioritizing interventions over the medium- and long-terms, hydraulic monitoring can also help operators track and improve supply schedules on a daily basis and identify acute operational failures that require immediate operator attention (see Section 3.3). Equipped with monitoring tools, operators will be able to respond more rapidly to failures in the system and provide a more predictable and reliable service.

We were not the first to install online monitoring stations in Arraiján. As part of a 2010 project, the network was divided into eight District Metering Areas (DMAs), and 15 pressure and flowrate monitoring stations, not altogether different from the ones used in this study, were installed at the boundaries of each DMA [26]. At the time of our research, however, only some of the sensors were working, the original telemetry equipment to upload the data to the internet was not working, and water balances for the DMAs had not recently been calculated.

Several factors could explain why the previously installed sensors were not maintained. A specialized division of the utility, located 20 km away in Panama City, was in charge of maintaining and collecting data from the Arraiján monitoring stations and many others throughout the country but had few logistical and human resources to dedicate to the task. The stations were set up by a private contractor, and the capacity to train utility personnel on how to use and maintain them may have been inadequate. Apart from these resource limitations, utility staff may not have seen sufficient reason to prioritize maintenance and use of the sensors when their resources were already stretched thin by immediate operational problems. With limited personnel available to analyze and use the monitoring data, and local operators not involved in that process or able to use the data as an operational tool, the data's value, and thus the value of maintaining the monitoring stations, may have been seen as low. If new tasks such as information collection are seen as add-ons that are not integral to existing tasks, compliance from frontline workers may be low [36].

Thus, as seen from previous experience in Arraiján, the type of monitoring recommended herein will only be useful if the utility has the human and logistical resources required to maintain sensors, analyze and interpret the data they produce, and take corrective actions based on the data. Promptly detecting a pipe break will be of little value if a repair crew is not available to fix it.

As mentioned in the introduction, hydraulic conditions are often spatially heterogeneous in intermittent systems, and monitoring an entire system like Arraiján's in detail may be costly. Monitoring costs could perhaps be reduced by rotating monitoring equipment around to different problem areas or developing inexpensive monitoring systems that provide only the most critical information required by operators and managers, such as whether the supply is off or on. Also, while the cost of extensive monitoring may seem high when viewed in isolation, it may still be small in comparison to a utility's capital improvements budget or operators' and users' coping costs associated with intermittent supply.

### 3.4.2. Making Irregular Supply More Predictable

An irregular supply that does not follow a schedule is inherently more unpredictable for users. To mitigate irregularity, users can be notified of variations in the normal schedule. During the time of this study, IDAAN was providing some such notification on a large scale by alerting and updating its customers about outages and repairs via a national Twitter feed. However, that broad approach may not be efficient for notifying customers of schedule changes in small areas like the study zones considered here. Other more targeted notification approaches may have the potential to make supply more predictable even when it is not regular. NextDrop, a start-up company, has attempted to achieve that in intermittent systems in India by notifying customers via text message when supply is about to be turned on by a valve operator, however achieving consistent compliance with the program from valve operators has proved challenging [20,36].

### 3.4.3. System-wide Strategies for More Continuous and Reliable Supply

As discussed in Section 3.3, some unexpected outages in intermittent supply areas and in areas normally receiving continuous supply were the result of widespread loss of supply or reduction of pressure brought on by operational failures such as treatment plant outages and transmission main breaks. While it will be impossible to completely prevent such operational failures, measures can be taken to reduce their incidence and mitigate their effects when they do occur.

An analysis of system-wide pipe breaks in Arraiján (Section 3.2) indicated that the system's very high break rate is driven mostly by certain areas that make up a small portion of the system. While system operators were very much aware that certain areas and certain pipes were the most prone to breaks, these chronic problems often went unaddressed. Documentation and analysis of pipe break data would likely help operators to keep utility managers and decision-makers better informed, and could motivate engineering and optimization studies to control pressure transients in problematic areas. While we focused on pipe breaks, better tracking and documentation of other operational failures such as treatment plant failures and power outages might help the utility to better understand and address their root causes as well.

It is also important to mitigate the effects of operational failures when they do inevitably occur. A robust distribution system, with sufficient storage capacity and a reserve supply is better able to cope with operational failures. The Arraiján system had plenty of supply but was not robust due to a lack of storage capacity and water loss rates so high that even the existing storage capacity could not be filled reliably. Accurate information on pipe connectivity and the availability of operable control valves to isolate areas where problems occur would also help limit the effects of failures. Reducing water losses, increasing storage capacity, and improving operators' ability to control the system would go a long way in helping the utility to mitigate the effects of operational failures.

### 3.5. Applicability of Results to Improvement of IWS

Much has been written about the risks and challenges posed by IWS [3,6,12,15,16], and researchers have proposed innovative strategies to model [11,14,18,19,37,38] and optimize the planning and operation of IWS networks [11,12,38–43]. In the literature, these strategies are normally implemented theoretically for example distribution networks. To successfully implement such strategies in practice, it will be critical to consider not just the topology of the IWS networks where they are applied, but also the reality of how such networks are operated by humans and the supply that results under baseline conditions. Unfortunately, very few data have been published that characterize the supply conditions in the wide variety of IWS networks throughout the world.

This detailed account of operation and supply in four sectors of one distribution network in Arraiján, Panama is intended to help fill that void. By selecting four supply zones that each had a different supply situation, we aimed to capture a wide spectrum of IWS. We expect that there are commonalities between these supply zones and those in many other IWS networks, such that some of the findings and recommendations are transferable. Nevertheless, this is one study of one network, and more work is needed to document and better understand the wide variety of IWS situations throughout the world.

## 4. Conclusions

Continuous hydraulic monitoring in four distinct zones of the Arraiján distribution network revealed that supply continuity varied widely between the zones and also temporally within each zone. The supply schedule was often irregular and unreliable in the intermittent study zones and sometimes interrupted by infrastructure failures that were not corrected for several days. Such unreliability made intermittent supply, already an inconvenient situation, even worse for users, and exacerbated variability in service.

The direct and underlying causes of intermittent supply in each zone were described, and based on this understanding, opportunities are identified to improve supply reliability by addressing operational problems at the local and system-wide scale. Continuous pressure monitoring in intermittent supply sectors would alert operators to unexpected or prolonged supply interruptions, allow supervisors to monitor the supply schedule received by users, and provide data to help managers and planners prioritize infrastructure investments and optimization efforts. On a larger scale, reducing water losses and providing adequate distribution system storage would make the network more robust to operational failures that sometimes cannot be avoided.

The proposed strategies (with the exception of increasing distribution storage) are centered on monitoring, data analysis, and gradual operational improvements rather than large capital infrastructure investments. Such improvements will require investments in the utility's human capital–its staff, and the resources they need to do their jobs effectively.

**Supplementary Materials:** The following are available online at http://www.mdpi.com/2073-4441/12/8/2143/s1, Figure S1: Location of the study zones and upstream monitoring points, Figure S2: Schematic of Study Zone 1, Figure S3: Schematic of Study Zone 3, Figure S4: Schematic of Study Zone 4, Figure S5: Pressure transient at 3:30 a.m. 18 November 2014 at the discharge of the Zone 4 pump station caused by the startup of the second of two pumps, Table S1: The ten zones with the highest break rates.

**Author Contributions:** All three authors participated in the conceptualization, methodology design, and funding and resource acquisition for this research. J.J.E. conducted most of the investigation and analysis, with contributions from K.L.N. and Y.C.Q. J.J.E. led the writing of the manuscript with continual input from K.L.N. All authors reviewed and edited article. All authors have read and agreed to the published version of the manuscript.

**Funding:** This work was funded by the Inter-American Development Bank (IADB, RG-T2441), a USAID Research and Innovation Fellowship, an NSF Graduate Fellowship, the Blum Center for Developing Economies and Henry Wheeler Center for Emerging and Neglected Diseases at UC Berkeley.

**Acknowledgments:** Panama's National Institute of Aqueducts and Sewers (IDAAN) contributed valuable staff time and in-kind resources for the project. Pipe break data were compiled by Javier Agrazal. We are grateful for assistance from Alejandra Perroni, Stefan Buss and Gustavo Martinez at IADB; Mauro Romero and many others at IDAAN; Carlos I. González; Paul West; and Joshua Kennedy.

**Conflicts of Interest:** The authors declare no conflict of interest. IADB staff provided input into the methodology development and interpretation of results, but did not participate in the drafting of this article or the decision to publish it.

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
