# Peer review of "Characterizing Supply Variability and Operational Challenges in an Intermittent Water Distribution Network"

_water, doi:10.3390/w12082143_

Round 1

Reviewer 1 Report

I've read the paper Entitled "Characterizing supply variability and operational challenges in an intermittent water distribution network" and my suggestion is to reject the paper.

The main reason are:

1) The paper looks more like an internal report of a managing authority of the water supply system or a consulting work rather than a scientific paper;

2) The paper lacks of scientific novelties: describing a monitoring system and its some data resulting from the observations of a speific case of study does not add any knowledge to the subject. 

Author Response

Point 1: The paper looks more like an internal report of a managing authority of the water supply system or a consulting work rather than a scientific paper;

Point 2: The paper lacks of scientific novelties: describing a monitoring system and its some data resulting from the observations of a specific case of study does not add any knowledge to the subject. 

Response to Points 1 and 2: The research presented is admittedly a case study with a significant focus on descriptive data. However, given the wide variety of IWS situations that exist, we believe that such case studies are needed to provide researchers and practitioners with detailed information collected using objective means (e.g., sensors) on the hydraulic conditions in IWS systems and on how IWS systems are operated. This type of information, on the operational realities in IWS systems, is sorely lacking in the literature, which limits the ability of other researchers and practitioners to study them and develop solutions.  This manuscript contributes to the literature by providing such data for four different zones of the Arraiján network and proposing strategies to better monitor and manage IWS. A new section, 3.5, has been added to provide discussion of these contributions.

Reviewer 2 Report

The manuscript concerns the important issue of the characterizing supply variability and operational challenges in an intermittent water distribution network. The following remarks should be referred to. How in practice the results of the presented work can be used? This should be discussed in the point concerning discussion of the results. How generalizable are the findings? Can they be applied to other areas? How dependent are they to specific characteristics of the region under examination? The novelty of the paper should be explicitly highlighted. Discussion is limited and should be somewhat deepened, focusing on the applicability of the proposed method to different locations throughout the world, to increase the visibility of the study, together with comparison with other reported literature methods.

The authors should give reasons why the presented approoach is chosen to analyse supply variability and operational challenges in an intermittent water distribution network.

Author Response

We thank the reviewer for their helpful comments and have revised the article to address them. Below are point-by-point responses.

Point 1: How in practice the results of the presented work can be used? This should be discussed in the point concerning discussion of the results.

Response to Point 1: Based on the research results, a number of strategies are proposed in Section 3.4 to improve service quality (no changes made). A new Section 3.5 (Lines 563-578) was also added to further explain how results from this case study can be used to inform other strategies for improving IWS.

Point 2: How generalizable are the findings? Can they be applied to other areas? How dependent are they to specific characteristics of the region under examination?

Response to Point 2: Lines 573-578 were added in Section 3.5 to discuss applicability of results from this study to other situations. Based on the authors’ knowledge of other water systems in Panama, the study zones are representative of many others in the country, and detailed results from this study can be used to develop more holistic solutions for other water systems for which much less information is available. Aspects of the current study are expected to be insightful for many IWS systems outside of Panama as well. In any case, the detailed descriptions of IWS that are provided allow other researchers and practitioners to determine whether their situations are similar enough to our study zones for our results and recommendations to be applicable.

Point 3: The novelty of the paper should be explicitly highlighted.

Response to Point 3: Specific discussion of novelty was added in two locations:

  • Lines 102-107: While previous studies of IWS networks have described supply schedules and inequities based on general observations or surveys of users [21–23], this study provides a much more detailed characterization based on a full year of continuous flow and pressure monitoring in four different study zones. This study also provides novel insights by linking the measured supply patterns to specific operational events and challenges, observed through extensive informal interactions with and interviews of system operators.”
  • Lines 570-573: “Unfortunately, very few data have been published that characterize the supply conditions in the wide variety of IWS networks throughout the world. This detailed account of operation and supply in four sectors of one distribution network in Arraiján, Panama is intended to help fill that void.”

Point 4: Discussion is limited and should be somewhat deepened, focusing on the applicability of the proposed method to different locations throughout the world, to increase the visibility of the study, together with comparison with other reported literature methods

Response to Point 4: Additional discussion has been added to address this comment at the following places:

  • Lines 335-338 (uneven service quality), added: “Previous research has identified uneven service quality as a common problem with IWS [3,11,12] and provided evidence of such inequities in IWS networks [22]. Our continuous monitoring data from Arraiján supports those findings and provides a more detailed picture of what uneven service quality can look like.”
  • Lines 340-357 (irregular and unreliable supply): This section was re-framed to use Galaitsi et al.’s proposed classification of IWS into “Predictable”, “Irregular” and “Unreliable” intermittency and comment on the usefulness and limitations of that classification. To be consistent with the terminology in Galaitsi et al, additional changes were made to some of the terminology throughout the paper (Lines 73-74, 119, 369, 488, 491, 498 & 582). For example, “irregular” was replaced with “unreliable.”
  • Lines 423-426 (correlation of IWS and pipe breaks), added: “Although the data did not show a clear-cut correlation between IWS and water main breaks, IWS could have been more strongly associated with service line breaks and leaks, which were not analyzed. A previous study in Cyprus [35] found that implementation of IWS caused a significant increase in the vulnerability to failure for household service lines but not for larger water mains.”
  • Lines 563-578, added: “Section 3.5 Applicability of results to improvement of IWS” discussing how results can be used by other practitioners and researchers and limitations for applying our results to other IWS networks.

Point 5: The authors should give reasons why the presented approach is chosen to analyze supply variability and operational challenges in an intermittent water distribution network.

Response to Point 5: Lines 128-132, added: “Pressure and flow sensors provided a more detailed, accurate, and objective characterization of supply than would be obtained by interviewing operators or users. Continuous monitoring over 1 year allowed supply variability and anomalies that may not occur frequently to be captured. Informal interviews with operators allowed us to connect the supply observed to the reality of operating a complex intermittent system.”

Reviewer 3 Report

The authors have presented a practical guide to working out deficiencies in water distribution networks, specifically places with intermittent supply.

A few sentences describing the main points from the authors' previous paper (Ref. 17) would be useful. 

The fonts for captions need to be consistent.

Author Response

We thank the reviewer for the helpful comments and have revised the article to address them. Below are point-by-point responses.

Point 1: A few sentences describing the main points from the authors' previous paper (Ref. 17) would be useful.

Response to Point 1: Added summary to Lines 93-97: Despite sustained low and negative pressures and water quality sometimes being degraded during the first-flush period (when supply first returned after an outage), random grab samples consistently had good quality. These results contrasted with results reported from a previous study in India, where water quality in intermittent zones was highly degraded [21], indicating that water quality may vary greatly among intermittent systems depending on the context.”

Point 2: The fonts for captions need to be consistent.

Response to Point 2: Fig. 3 caption font was changed (Line 200) so that all are consistent.